# Fetal macrosomia, fetal insulin, and insulin-like growth factor- 1 among neonates in Lagos, Nigeria: A case-control study

Olukayode O. Akinmola[1]◉, Babasola O. Okusanya◉[2]*, Gbenga Olorunfemi◉[3]◉, Henry C. Okpara[4]‡, Elaine C. Azinge[5]‡

**1** Department of Clinical Pathology, Lagos University Teaching Hospital, Lagos, Nigeria, **2** Department of Obstetrics and Gynaecology, College of Medicine, University of Lagos, Lagos, Nigeria, **3** Division of Epidemiology and Biostatistics, School of Public Health, University of Witwatersrand, Johannesburg, South Africa, **4** Department of Clinical Pathology, College of Medicine, Nnamdi Azikiwe University, Awka, Nigeria, **5** Department of Chemical Pathology, College of Medicine, University of Lagos, Lagos, Nigeria

◉ These authors contributed equally to this work.
‡ HCO and ECA also contributed equally to this work.
* babakusanya@yahoo.co.uk

**Data Availability Statement:** All relevant data are within the paper and its Supporting Information files.

**Funding:** The authors received no specific funding for this work.

## Abstract

### Purpose

Fetal macrosomia is associated with perinatal injuries. The purpose of this study was to assess the relationship between fetal insulin, insulin-like Growth factor-1(IGF-1), and macrosomia in a resource-limited setting.

### Method

This was a case-control study at tertiary and secondary health facilities in Lagos, Nigeria. One hundred and fifty mother-neonate pairs were recruited, and their socio-demographic and obstetric history was recorded. Fetal cord venous blood was collected at birth, and neonatal anthropometry was measured within 24hrs of life. Insulin and IGF-1 assay were measured with Enzyme-Linked Immunosorbent Assay (ELISA). Pearson's Chi-square was used to assess the association between categorical variables and macrosomia. Spearman's rank correlation of insulin, IGF-1, and fetal anthropometry was performed. Multivariable logistic regression was used to evaluate the association of insulin and IGF-1 with fetal birth weight. A statistically significant level was set at P-value < 0.05.

### Results

Macrosomic neonates had mean fetal weight, fetal length, and occipitofrontal circumference (OFC) of 4.15±0.26kg, 50.85±2.09cm and 36.35± 1.22cm respectively. The median Insulin (P = 0.023) and IGF-1 (P < 0.0001) were significantly higher among macrosomic neonates as compared to normal weight babies. Maternal BMI at birth (p = 0.003), neonate's gender (p < 0.001), fetal cord serum IGF-1 (p < 0.001) and insulin assay (P-value = 0.027) were significant predictors of fetal macrosomia. There was positive correlation between cord blood

**Competing interests:** The authors have declared that no competing interests exist.

IGF-1 and birth weight (r = 0.47, P-value < 0.001), fetal length (r = 0.30, P-value = 0.0002) and OFC (r = 0.37, P-value < 0.001).

## Conclusion

Among participating mother-neonate dyad, maternal BMI at birth, neonate's gender, and fetal cord serum IGF-1 and serum insulin are significantly associated with fetal macrosomia.

## Introduction

Fetal macrosomia is a significant cause of maternal and perinatal morbidity, especially in low- and middle-income countries (LMIC) [1]. Higher risk of caesarean sections, post-partum haemorrhage, birth asphyxia, neonatal trauma, and neonatal mortality are common macrosomia complications [1]. The future risk of metabolic syndrome and cardiovascular risks are linked with low birth weight, while non-diabetic obesity and cancer risks have been linked to macrosomia [2]. The negative impact of these chronic conditions on the health system and socio-economic life of LMICs is enormous. Hence, evidence of the pathophysiological processes of placental markers of fetal growth in LMICs, such as Nigeria, is required [3].

Fetal growth and development entail a complex interplay of the maternal, placenta, and fetal factors [4–6]. Pre-conceptional and antenatal maternal nutrition might impact the risk of metabolic disorders in their offspring in later life. The growth hormone-insulin-like growth factor (GH-IGF) axis is a crucial driver of fetal growth processes [3]. Maternal and fetal genetics and environmental factors contribute to birth weight in different proportions in populations [4–6]. Maternal factors, such as age, parity, weight, and gestational weight influence fetal weight [4]. Also, reports from the Avon Longitudinal Study of Pregnancy and Childhood (ALSPAC) have shown that fetal growth is influenced by fetal genes and maternal uterine-placental factors [5–7]. Therefore, an impairment of the genetic and environmental precursors of fetal growth can lead to rapid post-natal growth catch up that can cause childhood obesity and insulin resistance [5, 6].

The term "fetal macrosomia" refers to oversized fetuses. Although terms such as overweight, large for gestational age (LGA), and heavy-for-dates have been used [8, 9], none of these terms distinguishes the fetus with an abnormal body composition from normal. To make the distinction, Potter and Craig proposed the term macrosomia for fetuses with organ-weight disproportion in relation to body weight [9]. Fetal macrosomia may be symmetrical (proportionate) or asymmetrical (disproportionate). These categories are based on the ponderal index of > 2.8 (greater than 97th percentile), for asymmetric fetal macrosomia and a ponderal index of 2.2–2.8 (between the 10th and 90th percentile), for symmetric fetal macrosomia [10]. However, in this study for practical purposes macrosomia was defined as birth weight $\geq$ 4000 gram at term [11, 12].

Insulin is a peptide hormone synthesized by beta cells of the islets of Langerhans of the pancreas and is pivotal to regulating fat and carbohydrate metabolism in the human body. Gestational diabetes has been associated with macrosomia because of the roles of hyperinsulinism in macrosomia [13–15]. Conversely, some researchers have reported no association between cord blood insulin levels and macrosomia. Yet, insulin resistance, fetal hyperleptinaemia, and hypothalamic changes documented in the macrosomic neonates have been linked to adult life adverse outcomes [16, 17]. Insulin-like Growth Factors I and II (IGF-I; IGF-II) are polypeptides, structurally homologous to insulin, and share many biological activities [18]. Maternal

and fetal IGF-1 is believed to be essential mediators in fetal growth because of its r mitogenic and metabolic actions. Maternal IGF-1 and fetal IGF-1 have been associated with macrosomia, irrespective of maternal diabetes mellitus. Higher fetal cord serum levels of IGF-1 and insulin were reported in large for gestation neonates compared with neonates appropriate for gestational age [17], with a positive correlation with fetal birth weight and other anthropometric parameters birth, including in women with malaria infection in pregnancy in Nigeria [19–22]. The differences between Caucasian and Chinese neonates on the effect of IGF-1 and fetal weight suggests a genetic influence as a higher cord serum level of IGF-1 and a significant positive association of IGF-1 with neonatal birth weight and length in neonates was seen in Caucasians [23]. Like other resource-constraint regions, sub-Saharan Africa has a high prevalence of low birth-weight neonates, yet macrosomic fetuses' birth occurs. It is also uncertain what the contributions of IGF-1 and insulin are to fetal macrosomia in this setting. Establishing an association between IGF-1, insulin and macrosomia among Nigerian babies can aid further interventional research on the feto-maternal pathophysiology of the incidence of macrosomia and its attendant immediate maternal complications at birth and later life associated disease risks reductions. This study's objectives were to measure fetal insulin levels and IGF-1 in macrosomic fetuses delivered at term and determine the correlation of insulin and IGF-1 with fetal anthropometric variables at birth in Lagos State, Nigeria.

## Methods

This was a case-control study of mother-neonate pair at the obstetric units of four secondary and one tertiary health facilities in Lagos State. The secondary health facilities were General Hospital Surulere, General Hospital Isolo, General Hospital Mushin, Lagos Island Maternity, and the tertiary hospital was the Lagos University Teaching Hospital (LUTH), Lagos.

Mother-neonate pairs were consecutively recruited in the immediate postpartum period. Neonates were eligible if they were delivered at term (37 weeks to 42weeks gestational age). Cases were macrosomic babies (weight $\geq$ 4000g) (n = 100), while controls were AGA babies (n = 50). Exclusion criteria included multiple gestation, the presence of a fetal abnormality, and stillbirth.

Information on weight at the first antenatal visit, last normal menstrual period, weight at last antenatal visit, height, co-morbid states such as diabetes, hypertension, history of alcohol ingestion, parity, previous history of gestational diabetes mellitus, and previous history of macrosomic babies were obtained from the women and their medical records. The Body Mass Index (BMI) was then calculated as $Weight/(Height)^2$. Neonatal anthropometric measurements taken within 24hrs of birth by the research team were fetal length, occipitofrontal circumference, and birth weight (using SECA 813 flat digital weighing scale, Hamburg Germany).

Fetal cord blood was collected from the placenta's umbilical vein immediately after delivery with a 10ml syringe and placed into potassium EDTA bottles for IGF-1 and Insulin, and fluoride oxalate bottle for glucose assay. After temporary storage and transportation on Ice packs in a cooler, they were separated within a 2hrs of collection by centrifugation at 4,000 revolutions per min for 10mins to obtain the plasma. Glucose assay was performed within 6hrs of sample collection. The insulin and IGF-1 serum samples were made in aliquots into microtubes, batched, and stored in the central research laboratory at -80 $^0$C until the recruitment conclusion. Analysis of all the samples was then conducted together.

A commercially prepared immunoassay kit (Monobind Inc. Lake Forest, CA 92630, USA), using the Enzyme-Linked Immunosorbent Assay (ELISA) method, was used to measure cord blood insulin. IGF-1 was assayed using the ELISA technique for quantitative measurement of

Human IGF-1 in serum, plasma, and cell culture supernatant from Mediagnost GmbH, Aspenhaustr.25, 72770 Reutlingen, Germany (REF: E20, lot 120115). Glucose in plasma was assayed using the Trinder method. Spectrophotometry was used to measure the absorbance of the colored complex proportional to the concentration of glucose in the specimen measured at 500nm. Precision controls were used during the analysis. Within a run, within a batch, and day-to-day precision studies were carried out as required.

Approval was obtained from the Human Ethics Research Committee of the Lagos University Teaching Hospital (Ref No: ADM/DCST/HREC/APP/1810). The Heads of the General Hospitals provided permission for the study. Informed written consent was obtained from the mothers. The ethics committee approved the consent and all other aspects of the research. There was however no potential harm to the mothers and babies as this was an observational study with no intervention. Autonomy and confidentiality were maintained throughout the study.

## Statistical analysis

Stata version 16 (StataCorp, Texas, USA) statistical software was used for data analysis. Continuous variables were described using Mean ± Standard deviation (SD) or median and inter-quartile range (IQR)–if skewed data. Categorical variables were presented as frequencies and percentages. The mean of normally distributed variables was compared across the macrosomic and AGA babies using the student's independent t-test. Continuous variables that were not normally distributed were compared with the Mann-Whitney U test.

In contrast, Pearson's Chi-square (of Fisher's exact test for small numbers) was used to assess the association between categorical variables and the outcome status. Correlation of cord serum IGF-1 and Insulin with neonatal anthropometric variables (birth weight; occipito-frontal circumference; fetal length) was assessed using the Spearman's Rank correlation coefficient. Univariable and multivariable binary logistic regression was conducted to evaluate macrosomia's predictors with fetal IGF-1 as the primary explanatory variable. Variables with univariable P-value <0.2 was used to build the multivariable model using the backward elimination method. Some variables were chosen *a priori*. Similarly, two other multivariable models were built with serum insulin and glucose as the primary explanatory variable. Variables in the multivariable model I includes: IGF-1, maternal age, maternal BMI at birth, parity, gestational age, ethic group, and fetal sex. For models II and Model III, IGF-1 was replaced with insulin and glucose respectively A two-tailed test of the hypothesis was assumed, and Statistical significance was set at P-value <0.05 for all tests.

## Results

One hundred and fifty mother-neonate pairs were recruited into the study from 1st June to December 2015. They comprised 100 macrosomic (birth weight ≥ 4Kg) and 50 normal weight (Birth weight < 4kg) neonates. The mean maternal age was 31.6 ±4.6 years. The mean maternal pre-pregnancy and post-partum weight was 70.5 ± 12.7Kg and 86.9±14.4Kg, respectively. Most women (98%) did not have diabetes, though 33 (22%) women had a macrosomic baby in a previous birth. There was no statistically significant difference in the characteristics of mothers of macrosomic and normal-weight babies except for Parity, Body mass index, alcohol consumption, and Previous delivery of a macrocosmic baby. Other maternal characteristics are shown in Table 1. The proportion of males (n = 59/73, 80.8%) that were macrosomic was higher than the proportion of females that were macrosomic (n = 41/77, 53.3%), P-value <0.001. For macrosomic neonates, the mean fetal weight, fetal length, and occipitofrontal circumference (OFC) was 4.2±0.3kg, 50.8±2.1cm, and 36.3± 1.2cm, respectively, while the values

**Table 1. Socio-demographic characteristics of the participants.**

| Characteristics | Macrosomic babies N = 100 (%) | Normal weight babies N = 50 (%) | Total N = 100 (%) | P-value | Characteristics | Macrosomic babies N = 100 (%) | Normal weight babies N = 50 (%) | Total N = 100 (%) | P-value |
|---|---|---|---|---|---|---|---|---|---|
| Maternal Age Mean (SD) | 31.7 ± 4.6 | 32.4 ± 4.7 | 31.6 ± 4.6 | 0.6628 | **Pre-pregnancy BMI Mean (SD)** | 27.1 ±4. | 24.7± 5.1 | 26.3 ± 4.7 | 0.0031* |
| 20–24 | 6 (6.0) | 3(6.0) | 9 (6.0) | 0.974 | Underweight (<18.5) | 0 (0.0) | 2 (4.0) | 2 (1.3) | 0.003* |
| 25–29 | 26 (26.0) | 15 (30.0) | 41 (27.3) | | Normal weight (18.5–24.9) | 35 (35.0) | 30 (60.0) | 65 (43.3) | |
| 30–34 | 37 (37.0) | 16 (32.0) | 53 (35.3) | | Overweight (25.0–29.9) | 38 (38.0) | 12 (24.0) | 50 (33.3) | |
| 35–39 | 26 (26.0) | 13 (26.0) | 39 (26.0) | | Underweight (<18.5) | 0 (0.0) | 2 (4.0) | 2 (1.3) | 0.003* |
| 40–44 | 5 (5.0) | 3 (6.0) | 8 (5.3) | | **BMI at delivery Mean (SD)** | 33.8 ±4.6 | 29.7± 5.1 | 32.4 ± 5.1 | <0.001* |
| **Educational qualification** | | | | | Underweight (<18.5) | 0 (0.0) | 0 (0.0) | 0 (0.0) | < 0.001* |
| Primary | 2 (2.0) | 0 (0.0) | 2 (1.3) | 0.737 | Normal weight (18.5–24.9) | 3 (3.0) | 4 (8.0) | 7 (4.7) | |
| Secondary | 28 (28.0) | 16 (32.0) | 44 (29.3) | | Overweight (25.0–29.9) | 17 (17.0) | 24 (48.0) | 41 (27.3) | |
| Tertiary | 70 (70.0) | 34 (68.0) | 104 (69.3) | | **History of Hypertension** | | | | |
| **Parity** | 1.5 (1–2) | 0(0–2) | 1(0–2) | 0.0008* | Yes | 4 (4.0) | 1(2.00) | 5 (3.3) | 0.665 |
| 0 | 18 (18.0) | 25 (51.0) | 43 (28.9) | < 0.001* | No | 96 (96.0) | 49 (98.0) | 145 (96.7) | |
| **1–4** | 77 (77.0) | 23 (46.9) | 100 (67.1) | | **Alcohol Consumption** | | | | |
| **Parity** | 1.5 (1–2) | 0(0–2) | 1(0–2) | 0.0008* | Yes | 9 (9.0) | 0 (0.00) | 9 (6.0) | 0.030* |
| **History of smoking** | | | | | No | 91 (91.0) | 50 (100.0) | 141 (94.0) | |
| Yes | 0 (0.00) | 0 (0.00) | 0 (0.00) | | **History of Diabetes** | | | | |
| No | 100(100.0) | 100(100.0) | 100(100.0) | | Yes | 3 (3.0) | 0 (0.0) | 3 (2.0) | 0.551 |
| **Previous delivery of a macrocosmic baby** | | | | | No | 97 (97.0) | 50 (100.0) | 147 (98.0) | |
| No | 73 (73.0) | 44 (88.0) | 117 (78.0) | 0.039* | **Gestational age at delivery (weeks)** | 39.5 (39–40) | 39.5 (38–40) | 39.5 (38.3–40) | 0.1293 |
| Yes | 27 (27.0) | 6 (12.0) | 33 (22.0) | | **Birth weight** | 4.2±0.3 | 3.07± 0.3 | 3.8± 0.6 | < 0.001 |
| **Gender of the babies** | | | | | | | | | |
| Female | 41(41.0) | 36 (72.0) | 77 (51.3) | < 0.001 | | | | | |
| Male | 59 (59.0) | 14 (28.0) | 73 (48.7) | | | | | | |

SD: Standard deviation.

for normosomic neonates were 3.1±0.3kg, 47.9± 2.9cm, and 34.5±7.4cm respectively(P-value = 0.0001).

As shown in Fig 1, the median insulin levels were higher among macrosomic neonates, compared to the normosomic babies (4.84 (IQR: 3.1–10.1) IU/m Vs. 3.5 (2.8–6.4) IU/ml, P-value = 0.023). Similarly, the median IGF-1 level was higher among Macrosomic babies compared to Normosomic babies (85.1 (62.1–108.9) ng/ml Vs 53.8 (44.5–67.0) ng/ml, P-value < 0.0001). There was no statistically significant difference between the mean cord blood glucose levels of macrosomic and normosomic babies of 74.5 ± 30.9 mg/dl and 78.5±19.2 mg/dl, respectively (P-value = 0.4007).

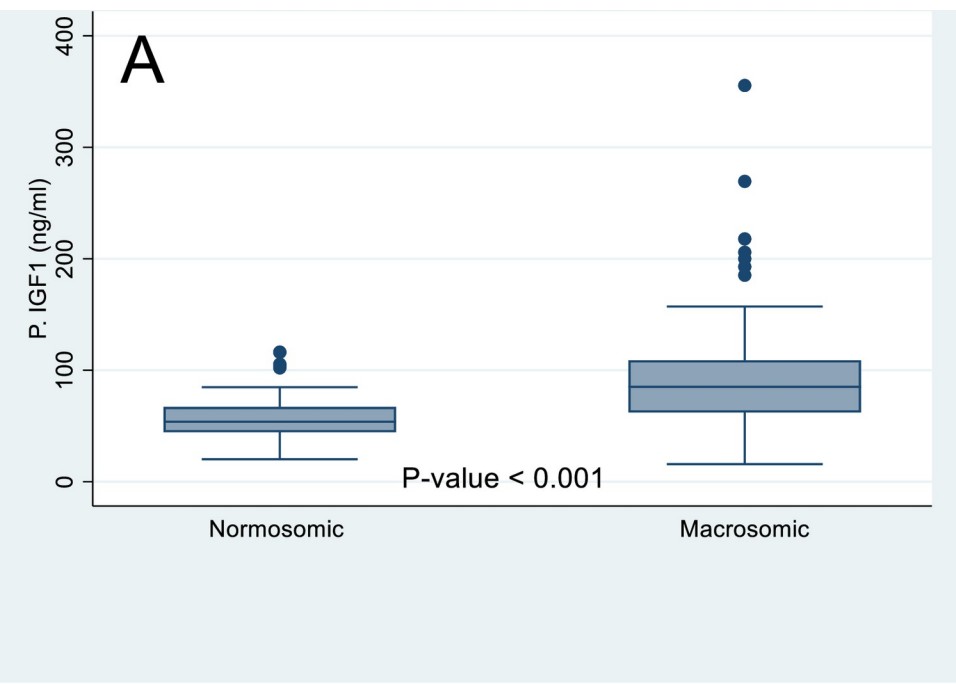

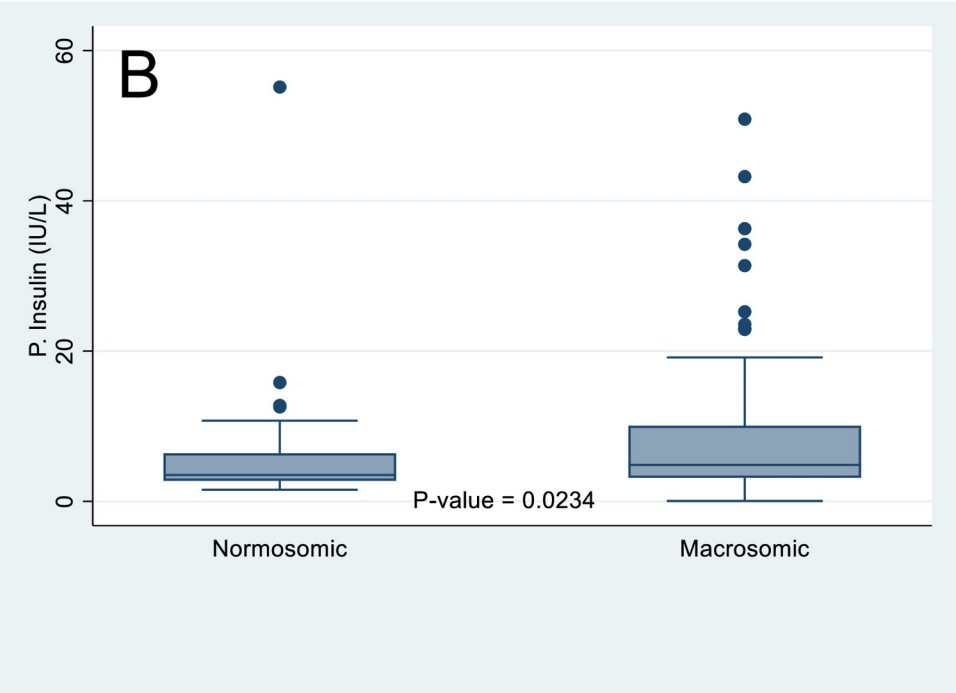

**Fig 1. A.** Distribution of GF-1(ng/ml) among macrosomic and normosomic babies. **B.** Distribution of Insulin (IU/L) among macrosomic and normosomic babies.

After multivariable analysis that corrected for confounding factors, maternal BMI at birth (P-value < 0.001), fetal gender (P-value = 0.001), fetal cord serum IGF -1 (P-value = 0.002), and insulin assay (P-value = 0.034) were significantly associated with fetal macrosomia. Also,

**Table 2. Logistic regression of the association between insulin, insulin-like growth factor glucose and fetal macrosomia.**

| Variables | Unadjusted Odds Ratio (OR) | 95%Confidence interval | P-value | Adjusted OR | Confidence Interval | P-value* |
|---|---|---|---|---|---|---|
| **Insulin assay** | 1.05 | 0.99–1.11 | 0.078 | 1.07 | 1.01–1.1 | 0.034* |
| **IGF-1** | 1.03 | 1.02–1.05 | < 0.001 | 1.03 | 1.01–1.05 | 0.002*£ |
| **Glucose levels** | 0.99 | 0.98 1.01 | 0.404 | 0.99 | 0.98–1.00 | 0.195¥ |
| **Maternal age** | 1.02 | 0.94–1.10 | 0.615 | 0.95 | 0.86–1.06 | 0.366 |
| **Pre-pregnancy Maternal BMI** | 1.13 | 1.04–1.23 | 0.004 | | | |
| **Maternal BMI at delivery** | 1.21 | 1.11–1.33 | < 0.001* | 1.22 | 1.10–1.35 | < 0.001* |
| **Parity** | 1.56 | 1.15–2.13 | 0.004 | 1.37 | 0.94–1.99 | 0.103 |
| **Gestational age at delivery (weeks)** | 1.31 | 0.95–1.79 | 0.098 | 1.53 | 1.02–2.30 | 0.039 |
| **Ethnic group** | | | | | | |
| Others | 1.00 | Reference | Reference | 1.00 | Reference | Reference |
| Yoruba | 0.37 | 0.08–1.81 | 0.220 | 0.73 | 0.12–4.37 | 0.728 |
| Igbo | 0.38 | 0.08–1.90 | 0.239 | 0.47 | 0.08–2.70 | 0.398 |
| **Fetal sex** | | | | | | |
| Female | 1.00 | Reference | Reference | 1.00 | Reference | Reference |
| Male | 3.70 | 1.77–7.72 | < 0.001* | 4.4804 | 1.85–10.82 | 0.001* |

£: The p-value was obtained from a different (second) model.

¥: The p-value was obtained from the third model.

* Statistically significant at P-value < 0.001.

Variables in the multivariable model I includes Insulin, maternal age, maternal BMI at birth, parity, gestational age, ethic group, and fetal sex.

For models II and Model III, insulin was replaced with IGF-1 and glucose respectively

for every unit increase in IGF-1, the odds of having a macrosomic baby increased by 3% (Adjusted OR: 1.03, 95%CI: 1.01–1.05, P-value = 0.002) Table 2.

Spearman's rank correlation of cord blood IGF-1 with neonatal anthropometry indicates positive correlation with birth weight (r = 0.47, P-value < 0.001), fetal length (r = 0.30, P-value = 0.0002) and OFC (r = 0.37, P-value < 0.001) (Fig 2). Similarly, cord blood Insulin had statistically significant positive correlation with fetal length (r = 0.19, P-value = 0.0201) and birth weight (r = 0.18, P-value = 0.0270) but there was no statistically significant relationship with OFC (r = 0.13; P-value = 0.1157). The relationship between glucose and the birth anthropometry, showed negative correlation and was not statistically significant for the Fetal length (r = -0.002; P-value = 0.98), OFC (r = -0.114; P-value = 0.16) and birth weight (r = -0.061; P-value = 0.46).

## Discussion

The study objectives were to measure fetal cord levels of IGF-1 and insulin and assess their correlation with the anthropometric indices of the neonates. Compared to normosomic neonates, insulin and IGF-1 were significantly higher in macrosomic neonates. Macrosomic neonates had median insulin and IGF-1 of 4.84IU/ml and 85.09 ng/ml, respectively. Maternal BMI at birth, fetal gender, and fetal cord serum IGF-1 were the significant predictors of fetal macrosomia on multivariable regression. Also, there was a weak positive correlation of cord blood IGF-1 with fetal anthropometric indices of birth weight, fetal length, and OFC. While cord blood Insulin significantly correlated with fetal length and birth weight, it had no statistically significant correlation with OFC.

Fetal growth regulation differs significantly from the postnatal growth process. While maternal and placenta factors, especially in the late trimester, influence fetal growth, childhood

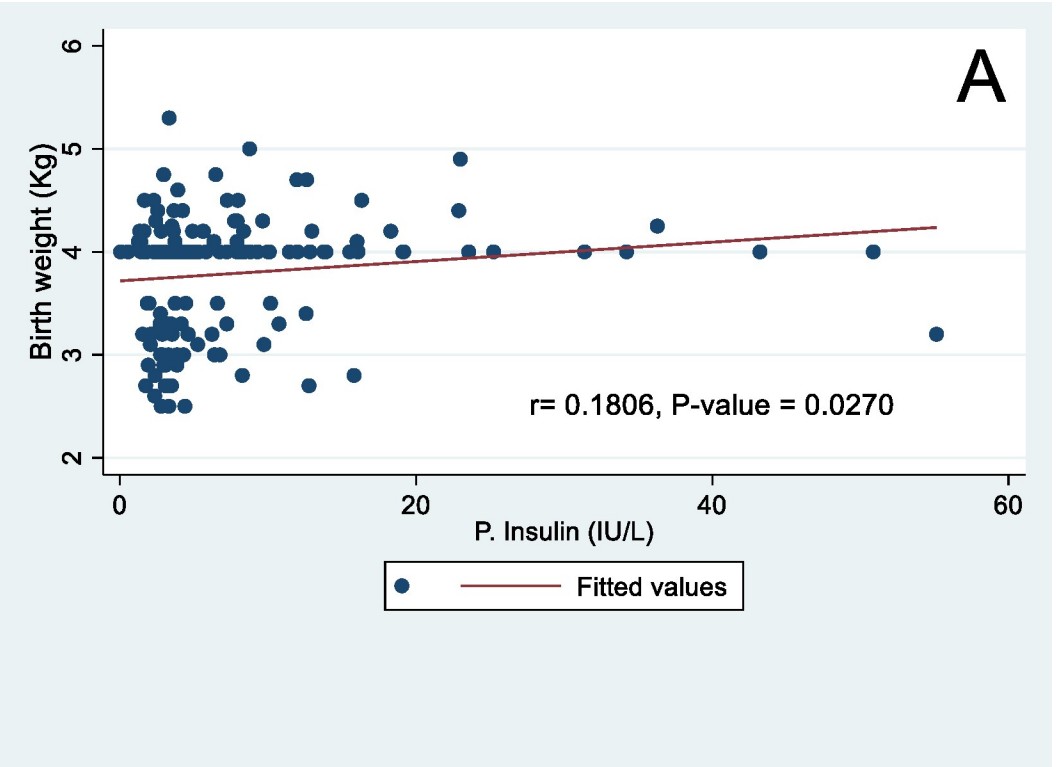

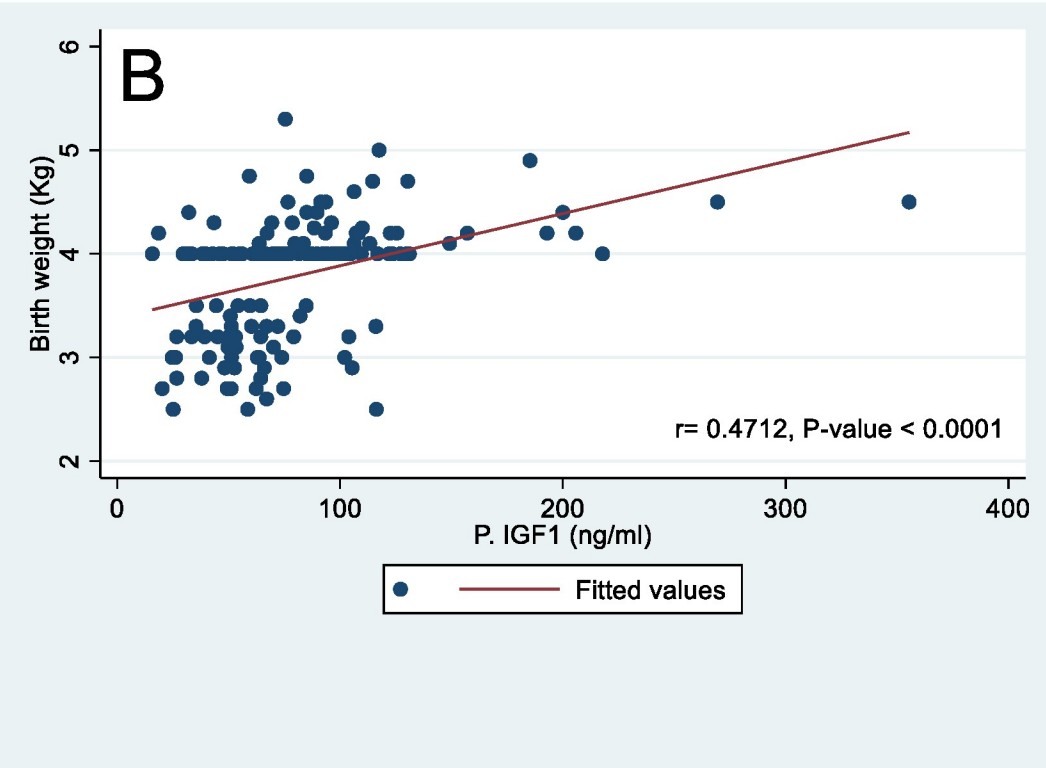

**Fig 2. Spearman's correlation coefficient of cord blood insulin and IGF-1 with birth weight.**

growth is exclusively influenced by genetic factors. This study reported a statistically significant association of fetal macrosomia with maternal parity, maternal height, previous history of big baby, and gender of the baby, with male dominance in macrosomia. Aside from maternal characteristics reported to influence birth weight, the neonates' gender has also been associated with macrosomia in previous studies [24–26].

Several authors have reported the role of insulin, IGF-1, and glucose in the determination of birth size, and by extension, fetal macrosomia in addition to fetal anthropometry [19, 20, 22]. Unlike the link of maternal hyperinsulinemia to excessive fetal growth as proposed by Pedersen's theory [27], most women in this study were euglycemic, including few diabetic women with good blood sugar control. Hence, maternal hyperglycemia with fetal hyperinsulinemia has little contribution to fetal macrosomia in this study. While Jacksic et al. reported the lower cord blood glucose levels in macrosomic babies due to higher insulin [26], Wizniter reported IGF-1 as the primary driver for growth because of higher insulin levels in macrosomic neonates [28]. However, the conclusions of Wizniter et al. were limited by the small sample size of the macrosomic study population [28]. In a sample of euglycemic mothers, insulin and IGF-1 levels were significantly higher in macrosomic neonates, with IGF-1 levels strongly associated with birth weight [26, 28, 29].

In this study, the level of IGF-1 was significantly higher in macrosomia compared to controls (85.09 vs. 53.94: p-value = 0.0001). A previous study had reported a strong positive linear relationship between cord IGF-1 birth weight and birth length [19]. Despite these reports, Chiesa et al. reported no significant difference in insulin levels of macrosomic and normosomic neonates [21], with authors attributing their findings to IGF-1 level's regulation by insulin. However, the study involved only 27 macrosomic babies [21].

This study reports a correlation of fetal length, head circumference, and birth weight with Insulin and IGF-1. Another study reported a correlation (r = 0.39 p<0.0001) between insulin levels and birth weight [30]. Evaluating the implications of malaria parasitemia on fetal anthropometric measures in a sample of Nigerian neonates, Ayoola et al. reported a significantly strong correlation of birth weight, insulin and IGF-1 [22]. However, there was no significant correlation between birth length, OFC, and Insulin among their cohort of babies. Similarly, their study did not reveal any correlation between birth length, OFC, and IGF-1. This difference between our results and those of Ayoola et al. might be due to anthropometric measurements of neonates whose mothers had malaria parasitemia [22].

This study provides the scientific basis for a relatively common clinical condition of fetal macrosomia. In Nigeria, fetal macrosomia incidence has been reported to be 5.5%- 8.1% [24, 31, 32]. It provides an association of IGF-1 and Insulin with fetal macrosomia in a largely non-diabetic population of pregnant women. The understanding that high IGF-1 and Insulin concentrations in fetuses are predispositions to fetal macrosomia might open a new vista of prenatal assessment of fetal IGF-1 and insulin as well as interventions to prevent fetal macrosomia and its associated maternal and fetal morbidities like higher caesarean section rates, operative vaginal deliveries, increased duration of birth, shoulder dystocia and genital lacerations [24, 31, 32].

Our study is the first in Nigeria to prospectively evaluate the relationship between IGF-1 and fetal macrosomia in euglycaemic mothers, thereby contributing to the growing evidence of the role of IGF-1 in growth and metabolic disorder in our environment. A strength of the study is the recruitment ration of 2:1 for cases and controls, unlike the usual 1:1 recruitment ratio. The collection of blood sample at birth, and fetal anthropometric measure within 24 hours of birth supports the results. We report a significant association between insulin and IGF-1 as a macrosomia marker, with glucose contributing little to fetal macrosomia, contrary to Pedersen's theory. The study limitations may include the recruitment of only euglycemic

mothers and the use of Hospital controls. Yet, this research's findings are like the reports of macrosomia in diabetic women. Due to the equivocation of the roles of hyperinsulinemia in fetal growth, there is the need to evaluate other fetal growth biologic fuel, with the hope of identifying the stimulus for the hyperinsulinaemic states of macrosomic babies.

## Conclusion

Fetal insulin and IGF-1 were significantly higher among macrosomic neonates compared to normosonic neonate. Among participating mother-neonate dyad, maternal BMI at birth, neonate's gender, and fetal cord serum IGF-1, and serum insulin were significantly associated with fetal macrosomia.

## Supporting information

**S1 Data.**
(XLS)

## Acknowledgments

We thank all the mothers and nurses that participated in this study.

## Author Contributions

**Conceptualization:** Olukayode O. Akinmola, Babasola O. Okusanya, Gbenga Olorunfemi, Henry C. Okpara, Elaine C. Azinge.

**Data curation:** Olukayode O. Akinmola, Babasola O. Okusanya.

**Formal analysis:** Olukayode O. Akinmola, Gbenga Olorunfemi.

**Funding acquisition:** Olukayode O. Akinmola.

**Investigation:** Olukayode O. Akinmola, Babasola O. Okusanya.

**Methodology:** Olukayode O. Akinmola, Babasola O. Okusanya, Gbenga Olorunfemi, Henry C. Okpara, Elaine C. Azinge.

**Project administration:** Olukayode O. Akinmola, Babasola O. Okusanya, Elaine C. Azinge.

**Resources:** Olukayode O. Akinmola.

**Software:** Gbenga Olorunfemi.

**Supervision:** Babasola O. Okusanya, Elaine C. Azinge.

**Validation:** Olukayode O. Akinmola, Babasola O. Okusanya.

**Visualization:** Olukayode O. Akinmola, Gbenga Olorunfemi.

**Writing – original draft:** Olukayode O. Akinmola, Babasola O. Okusanya.

**Writing – review & editing:** Olukayode O. Akinmola, Gbenga Olorunfemi, Henry C. Okpara, Elaine C. Azinge.

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
