## [Decision Letter · Decision Letter 0]

27 Oct 2021

PONE-D-21-16686Fetal macrosomia, fetal Insulin, and Insulin-like growth factor- 1 among neonates in Lagos, Nigeria: A case-control study

PLOS ONE

Dear Dr. Okusanya,

Thank you for submitting your manuscript to PLOS ONE. After careful consideration, we feel that it has merit but does not fully meet PLOS ONE’s publication criteria as it currently stands. Therefore, we invite you to submit a revised version of the manuscript that addresses the points raised during the review process.

The manuscript has been evaluated by one reviewer, and their comments are available below. The reviewer has raised a number of concerns that need attention. They request additional information on methodological aspects of the study and the interpretation of the results. I recommend that you pay particular attention to improving the discussion and justification of the scientific rationale of your study. Could you please revise the manuscript to carefully address the concerns raised?

We look forward to receiving your revised manuscript.

Kind regards,

Dario Ummarino, Ph.D.

Senior Editor

PLOS ONE

Journal Requirements:

2. Please amend your current ethics statement to address the following concerns: Please explain why written consent was not obtained, how you recorded/documented participant consent, and if the ethics committees/IRBs approved this consent procedure.

4. Please amend the manuscript submission data (via Edit Submission) to include author Oluwakayode O Akinmola.

5. Please amend your authorship list in your manuscript file to include author Kayode Akinmola.

6. We note you have included a table to which you do not refer in the text of your manuscript. Please ensure that you refer to Table 2 in your text; if accepted, production will need this reference to link the reader to the Table.

Reviewers' comments:

Reviewer's Responses to Questions

**Comments to the Author**

1. Is the manuscript technically sound, and do the data support the conclusions?

Reviewer #1: No

2. Has the statistical analysis been performed appropriately and rigorously? 

Reviewer #1: Yes

3. Have the authors made all data underlying the findings in their manuscript fully available?

Reviewer #1: No

4. Is the manuscript presented in an intelligible fashion and written in standard English?

Reviewer #1: No

5. Review Comments to the Author

Reviewer #1: General comments

It is difficult to understand at the time of reading what the rationale of the study is and why it seems so important in the Nigerian population to know if there is an association between fetal insulin and IGF1 levels and macrosomia. What clinical impact do the authors expect from their findings, if any?

Moreover, there is a discrepancy between the announced objective (relationship between fetal insulin, IGF1 and macrosomia) and the results and the conclusion which essentially concerns the maternal, neonatal and biological clinical predictive factors of macrosomia. Why is the study and the collected samples from 2015 and only submitted for publication in 2021?

Specific comments

MM

Justify the choice of a 2/1 control case study?

Why did you not match the controls on BMI and parity, which are major confounding factors?

what is the rationale for the evaluation of neonatal anthropometric measurements of the choice of occipitofrontal circumference ? It does not seem that it is a representation of the neonatal fat mass. What is Digital weighing scale ?

The definition of macrosomia is not given and if it is a question of babies weighing more than 4000g as it seems to appear in the results, then some babies are not macrosome since this depends on the gestational age at birth and therefore on the weight curves at birth. Indeed at term 4000kg is not a fetus > 90th percentile and is not considered as macrosomic newborn. Please give the definition of macrosomia and justify this choice for the interest of the study.

Table 1 should be simplified and fit on one page. On the other hand, as indicated, the groups between macrocosmic and non-macrocosmic fetuses are not comparable and BMI and parity appear as confounding factors and limit the interpretation of the results.

Results

Line 204 It is not indicated either in MM or in the results which confounding factors were entered in the logistic regression model. Please indicate these details

When interpreting the results in Table 2, the sentence maternal BMI at birth (P205

value =0.003), fetal gender (P-value < 0.001), fetal cord serum PGF-1 (< 0.001), and insulin

206 assay (P-value =0.027) were the statistically significant predictors of fetal macrosomia should be changed to "were significantly associated of ..." This is not a prediction model

Similarly, in the conclusion, the word prediction should be changed to association.

Discussion

It seems when reading the discussion that it has been reported in non-diabetic macrocosmic fetuses an increase in insulin and IGF1 and this study just reports the same results but in a Nigerian population. I have difficulty understanding the clinical impact of these results and this is not at all discussed. Metabolic consequences in the macrosomic newborn? Particular follow-up of these children?

There are many points to review in the objective, methodology and results in order to better understand the message that the authors wish to give in this article

6. PLOS authors have the option to publish the peer review history of their article (what does this mean?). If published, this will include your full peer review and any attached files.

Reviewer #1: No

---

## [Author Response · Author response to Decision Letter 0]

10 Dec 2021

6th December 2021

The Editorial Board

PlosOne

Thank you for a timely review of the manuscript. The authors have reviewed the comments/ edits of the reviewers and have made changes as suggested by them. Where it was impossible to make the suggested review, we have provided reasons for our position. For ease of identification, we have provided files with tracked changes and clean copy. Kindly find below a point-by-point response to the reviewers’ comments.

Response: We have ensured the compliance with the author’s guideline. However, if there we have omitted any requirement, it will promptly be attended to during production, if the manuscript is accepted for publication.

2. Please amend your current ethics statement to address the following concerns: Please explain why written consent was not obtained, how you recorded/documented participant consent, and if the ethics committees/IRBs approved this consent procedure.

Response: We obtained ethical approval for written consent, which was the consent procedure for this research. We have now indicated we obtained written consent in the manuscript as: “Informed written consent was obtained from the mothers. The ethics committee approved the consent and all other aspects of the research. There was however no potential harm to the mothers and babies as this was an observational study with no intervention. Autonomy and confidentiality was maintained throughout the study” line 169 – 173 (page 7)

Kindly note that informed written consent was obtained from the participants and was originally stated in the research dissertation at page 34 as published by the National Postgraduate Medical College of Nigeria. 1089-Article Text-6613-1-10-20190415 (3).pdf

Response: We have uploaded the data supporting our manuscript as supplementary file named S1.

3. Please amend the manuscript submission data (via Edit Submission) to include author Oluwakayode O Akinmola.

Response: This has been done. The correct name of Olukayode Akinmola has been added.

4. Please amend your authorship list in your manuscript file to include author Kayode Akinmola.

Response: This has been done. The correct name of Olukayode Akinmola has been added

5. We note you have included a table to which you do not refer in the text of your manuscript. Please ensure that you refer to Table 2 in your text; if accepted, production will need this reference to link the reader to the Table.

Response: This has now been done. Kindly refer to line 240, page 16. 

Reviewers' comments:

Reviewer's Responses to Questions

Comments to the Author

1. Is the manuscript technically sound, and do the data support the conclusions?

Reviewer #1: No

2. Has the statistical analysis been performed appropriately and rigorously?

Reviewer #1: Yes

3. Have the authors made all data underlying the findings in their manuscript fully available?

Reviewer #1: No

4. Is the manuscript presented in an intelligible fashion and written in standard English?

Reviewer #1: No

5. Review Comments to the Author

Reviewer #1: General comments

It is difficult to understand at the time of reading what the rationale of the study is and why it seems so important in the Nigerian population to know if there is an association between fetal insulin and IGF1 levels and macrosomia. 

Response: We have recast the manuscript and provided further study justification. We stated that the two main pathways of influence of fetal weight was fetoplacental and genetic factors. One of the feto-placental factors is the elaboration of insulin and IGF-1. In most paragraphs of the introduction section, the study justification was presented. Specifically, and in line with the reviewer’s comment, we stated:

“Establishing an association between IGF-1, insulin and macrosomia among Nigerian babies can aid further interventional research on the feto-maternal pathophysiology of macrosomia and its attendant immediate maternal complications at birth and later life associated disease risks reductions” line 123- 126 (Page 5) 

What clinical impact do the authors expect from their findings, if any?

Moreover, there is a discrepancy between the announced objective (relationship between fetal insulin, IGF1 and macrosomia) and the results and the conclusion which essentially concerns the maternal, neonatal and biological clinical predictive factors of macrosomia. 

Response: The authors did not see any discrepancies between the research objectives, results and conclusion. We reported on our primary objective of the association between IGF-1 and macrosomia. We further conducted multivariable regression modelling to assess the association between IGF-1 and macrosomia. We then reported other secondary findings. 

Why is the study and the collected samples from 2015 and only submitted for publication in 2021?

Response: the observed interval between research and publication is correct. We believe in transparent scientific reporting, which was why the dates are as reported. Essentially, the study was conducted in part fulfilment for the requirement of the award of the Fellowship of the National Postgraduate Medical College of Nigeria. It is being reported in a scientific journal now because the candidate’s mentors have encouraged him to disseminate his research findings in a peer-reviewed journal. 

Specific comments

MM

Justify the choice of a 2/1 control case study?

Response: Case control studies might have case: control ratio of 1:1, 1:2, 1:3, 1:4 or vice versa. We adopted a 2:1 ratio by recruiting 100 cases and 50 control to minimize the challenges of enrolment of controls, and to reduce the study duration and costs. Furthermore, some sub-analyses were conducted among the cases. Thus, a larger sample of the cases will improve the conclusion of such intraclass analyses.

Why did you not match the controls on BMI and parity, which are major confounding factors?

Response: Known confounders in research may be handled during participants recruitment and data analysis. Due to the difficulty of matching using more than one variable, matching was done with only one variable. However, we utilized multivariable regression modelling to correct for other known confounding variables, including parity. We have now added parity as a variable in the multivariable model (Table 2)

what is the rationale for the evaluation of neonatal anthropometric measurements of the choice of occipitofrontal circumference ? It does not seem that it is a representation of the neonatal fat mass. 

Response: We thank you for this question. We decided to show all the anthropometric measurements, including the occipitofrontal circumference. Although, at the moment OFC does not have biologic direct correlation with the neonatal fat mass, we felt that showing all the parameters might be valuable to the literature in future. If the reviewers and indeed the editor strongly feel otherwise, the authors will remove OFC as an anthropometric measure.

What is Digital weighing scale ?

Response: The weighing scale used for participants weight measurement has been further described. Line 147 - 148

The definition of macrosomia is not given and if it is a question of babies weighing more than 4000g as it seems to appear in the results, then some babies are not macrosome since this depends on the gestational age at birth and therefore on the weight curves at birth. Indeed at term 4000kg is not a fetus > 90th percentile and is not considered as macrosomic newborn. Please give the definition of macrosomia and justify this choice for the interest of the study.

Response: For this study, the definition of macrosomia is birth weight weight ≥ 4000g. This has been defined in the introduction (line 102 - 102) and methodology (line 138) with references. Also, we used an absolute weight of 4000g at birth as it is more pragmatic and will avoid the use of different cut-off weights for different neonates.

Table 1 should be simplified and fit on one page.

Response: The Table has been re-designed to fit 2-pages in Landscape format as it was not possible to make it a page.

On the other hand, as indicated, the groups between macrocosmic and non-macrocosmic fetuses are not comparable and BMI and parity appear as confounding factors and limit the interpretation of the results.

Response: Table 1 is a bivariate analysis. We corrected for the confounding variables in Table 2. Table 2 showed the univariable and multivariable analysis that corrected for BMI and parity. We re-analyzed the multivariable model by including parity in the model. This did not change the point estimates or direction of association of the relationship between insulin and the outcome. 

Results

Line 204 It is not indicated either in MM or in the results which confounding factors were entered in the logistic regression model. Please indicate these details

Response: Thank you for this observation. This has now been stated in the methods (line 192 - 194) and as legend of Table 2 

When interpreting the results in Table 2, the sentence maternal BMI at birth (P205

value =0.003), fetal gender (P-value < 0.001), fetal cord serum PGF-1 (< 0.001), and insulin

206 assay (P-value =0.027) were the statistically significant predictors of fetal macrosomia should be changed to "were significantly associated of ..." This is not a prediction model

Response: This has been corrected (Line 237 -238)

Similarly, in the conclusion, the word prediction should be changed to association.

Response: This has been corrected in the conclusion of the abstract and the conclusion after discussion (Line 331 -332)

Discussion

It seems when reading the discussion that it has been reported in non-diabetic macrocosmic fetuses an increase in insulin and IGF1 and this study just reports the same results but in a Nigerian population. I have difficulty understanding the clinical impact of these results and this is not at all discussed. Metabolic consequences in the macrosomic newborn? Particular follow-up of these children?

There are many points to review in the objective, methodology and results in order to better understand the message that the authors wish to give in this article

Response: This is a basic science research that provides the biochemical basis for fetal macrosomia, a relative common pregnancy complication in Nigeria. On Page 20, lines 308 - 315 of the revised manuscript, the authors have included clinical implications of the study findings, including a potential use of prenatal assay of fetal IGF-1 and insulin to predict fetuses who might develop fetal macrosomia. 

6. PLOS authors have the option to publish the peer review history of their article (what does this mean?). If published, this will include your full peer review and any attached files.

Do you want your identity to be public for this peer review? For information about this choice, including consent withdrawal, please see our Privacy Policy.

Reviewer #1: No

---

## [Editor Report · Decision Letter 1]

24 Jan 2022

PONE-D-21-16686R1Fetal macrosomia, fetal Insulin, and Insulin-like growth factor- 1 among neonates in Lagos, Nigeria: A case-control studyPLOS ONE

Dear Dr. Okusanya,

Thank you for submitting your manuscript to PLOS ONE. After careful consideration, we feel that it has merit but does not fully meet PLOS ONE’s publication criteria as it currently stands. Therefore, we invite you to submit a revised version of the manuscript that addresses the points raised during the review process.

We look forward to receiving your revised manuscript.

Kind regards,

Haim Werner

Academic Editor

PLOS ONE

Additional Editor Comments:

The aim of the present manuscript by Akinmola et al was to measure fetal insulin and insulin-like growth factor-1 (IGF-1) levels in macrosomic fetuses delivered at term in Lagos State, Nigeria. Overall, the research was aimed at determining potential correlations between both hormones with fetal anthropometric variables. Authors provide evidence that maternal BMI at birth, neonate’s gender, and fetal cord serum IGF-1 and insulin were correlated with fetal macrosomia. Authors emphasize that this association might help identify these hormones as important predictive factors to prevent macrosomia. Hence, the paper might bear important clinical relevance.

The present manuscript constitutes a revised version of the paper. Authors have addressed reviewer’s concerns in a satisfactory manner. In particular, authors expand on the rationale of the study and better elaborate the justification for the study. In addition, authors clarify a number of methodological issues in the revised version.

In summary, the present manuscript is a concise and straightforward analysis of the link between neonate insulin, IGF-1 and macrosomia. While mainly of a descriptive nature, the paper includes valuable informmation.

Minor points:

Lines 50, 203, 207, 238, 300, : should be IGF-I, not PGF-1.
---

## [Author Response · Author response to Decision Letter 1]

17 Mar 2022

Department of Obstetrics and Gynaecology,

College of Medicine, 

University of Lagos, 

Lagos, Nigeria.

16th March, 2022

The Editorial in Chief

PlosOne

Dear Editor-in-Chief,

Revised manuscript Titled Fetal macrosomia, fetal Insulin, and Insulin-like growth factor- 1 among neonates in Lagos, Nigeria: A case-control study 

We are grateful for the favorable outcome of our above titled manuscript. We have further edited the manuscript in line with the recommendations of the editor. Kindly find below a point-by-point response to the reviewers’ comments. We have uploaded a clean copy and a manuscript with tracked changes.

Editor’s comment:

The aim of the present manuscript by Akinmola et al was to measure fetal insulin and insulin-like growth factor-1 (IGF-1) levels in macrosomic fetuses delivered at term in Lagos State, Nigeria. Overall, the research was aimed at determining potential correlations between both hormones with fetal anthropometric variables. Authors provide evidence that maternal BMI at birth, neonate’s gender, and fetal cord serum IGF-1 and insulin were correlated with fetal macrosomia. Authors emphasize that this association might help identify these hormones as important predictive factors to prevent macrosomia. Hence, the paper might bear important clinical relevance.

The present manuscript constitutes a revised version of the paper. Authors have addressed reviewer’s concerns in a satisfactory manner. In particular, authors expand on the rationale of the study and better elaborate the justification for the study. In addition, authors clarify a number of methodological issues in the revised version.

In summary, the present manuscript is a concise and straightforward analysis of the link between neonate insulin, IGF-1 and macrosomia. While mainly of a descriptive nature, the paper includes valuable informmation.

Authors’ response: We are grateful for the favourable and fair assessment of our manuscript

Editor’s comment:

Minor points:

Lines 50, 203, 207, 238, 300, : should be IGF-I, not PGF-1.

Authors’ response:

We have changed PGF-1 to IGF-1 throughout the manuscript. We further utilized the “find” in Microsoft word to ensure all PGF-1 has been changed to IGF-1. 

Once again, we are grateful for the opportunity to publish in your prestigious journal

Yours faithfully,

Dr Babasola Okusanya (on behalf of co-authors)

---

## [Editor Report · Decision Letter 2]

21 Mar 2022

Fetal macrosomia, fetal Insulin, and Insulin-like growth factor- 1 among neonates in Lagos, Nigeria: A case-control study

PONE-D-21-16686R2

Dear Dr. Okusanya,

We’re pleased to inform you that your manuscript has been judged scientifically suitable for publication and will be formally accepted for publication once it meets all outstanding technical requirements.

Kind regards,

Haim Werner

Academic Editor

PLOS ONE

Additional Editor Comments (optional):

Authors have satisfactorily addressed reviewer's comments.
---

## [Editor Report · Acceptance letter]

18 Apr 2022

PONE-D-21-16686R2 

Fetal macrosomia, fetal Insulin, and Insulin-like growth factor- 1 among neonates in Lagos, Nigeria: A case-control study 

Dear Dr. Okusanya:

I'm pleased to inform you that your manuscript has been deemed suitable for publication in PLOS ONE. Congratulations! Your manuscript is now with our production department. 

Kind regards, 

on behalf of

Dr. Haim Werner 

Academic Editor

PLOS ONE